# Environmental Health Surveillance System for a Population Using Advanced Exposure Assessment

**DOI:** 10.3390/toxics8030074

**Published:** 2020-09-18

**Authors:** Wonho Yang, Jinhyeon Park, Mansu Cho, Cheolmin Lee, Jeongil Lee, Chaekwan Lee

**Affiliations:** 1Department of Occupational Health, Daegu Catholic University, Gyeongsan-si, Gyeongbuk 38430, Korea; s100002@naver.com; 2Center of Environmental Health Monitoring, Daegu Catholic University, Gyeongsan-si, Gyeongbuk 38430, Korea; venza11@naver.com; 3Department of Nano and Biological Engineering, Seo Kyeong University, Seoul 02713, Korea; cheolmin@skuniv.ac.kr; 4Korea Testing and Research Institute, Gwacheon-si, Gyeonggi-do 13810, Korea; emjilee@ktr.or.kr; 5Department of Occupational and Environmental Medicine, Busan Paik Hospital and Institute of Environmental and Occupational Medicine, Inje University, Busan 47392, Korea; lck3303@hanmail.net

**Keywords:** air pollution, exposure assessment, population exposure, environmental health surveillance system

## Abstract

Human exposure to air pollution is a major public health concern. Environmental policymakers have been implementing various strategies to reduce exposure, including the 10th-day-no-driving system. To assess exposure of an entire population of a community in a highly polluted area, pollutant concentrations in microenvironments and population time–activity patterns are required. To date, population exposure to air pollutants has been assessed using air monitoring data from fixed atmospheric monitoring stations, atmospheric dispersion modeling, or spatial interpolation techniques for pollutant concentrations. This is coupled with census data, administrative registers, and data on the patterns of the time-based activities at the individual scale. Recent technologies such as sensors, the Internet of Things (IoT), communications technology, and artificial intelligence enable the accurate evaluation of air pollution exposure for a population in an environmental health context. In this study, the latest trends in published papers on the assessment of population exposure to air pollution were reviewed. Subsequently, this study proposes a methodology that will enable policymakers to develop an environmental health surveillance system that evaluates the distribution of air pollution exposure for a population within a target area and establish countermeasures based on advanced exposure assessment.

## 1. Introduction

Air pollution poses a major environmental health problem and remains one of the biggest challenges faced by many countries. The World Health Organization (WHO) has reported that more than 80% of urban populations are exposed to air quality levels exceeding health guidelines [1]. Air pollutants originate from various sources, such as industry, transportation, and households. There is evidence demonstrating that exposure to air pollutants may increase the risk of diseases such as lung cancer and respiratory illnesses [2,3,4].

Previous research has largely focused on a range of air pollutants, including particulate matter (PM), ozone (O_3_), nitrogen dioxide (NO_2_), carbon monoxide (CO), volatile organic compounds (VOCs), sulfur dioxide (SO_2_), carbon dioxide (CO_2_), and benzene (C_6_H_6_) [5,6]. Although most air pollutants have been monitored, PM has been the most widely studied in terms of its environmental health risk and health effects; health risks from exposure to PM (PM_10_ and PM_2.5_) are more serious than those posed by other air pollutants [7]. Exposure to PM has been reported to have adverse respiratory and cardiovascular health effects, including decreased lung function, asthma, cardiopulmonary disease, and lung cancer [4,8,9].

Many studies on the health effects of air pollution exposure have used monitoring data from fixed air monitoring stations. As such, they lack variabilities in terms of spatio-temporal distribution and exposure within a population, leading to erroneous estimates [10]. Although the accuracy and precision of air pollutant measurements are high, as fixed air monitoring stations mainly use expensive international standard instruments, they are limited in their ability to assess exposure at the individual and population scales [11].

Air pollution exposure assessments for a population are required to evaluate the effectiveness of air quality control policies and implement effective interventions [12]. Exposure estimation for an entire population is dependent on the specific objectives of a study. However, it is important to integrate indoor and outdoor air pollution levels and individual time–activity patterns to generate reliable exposure assessments [13]. Recent studies have used new methodologies to assess exposure for the urban population based on alternative technologies and mobile monitoring approaches [14,15,16]. However, there is still a need to improve the air pollution exposure assessment for a population. For example, the patterns of time-based activities at the individual scale were collected through census data, administrative registers, time–activity surveys, and existing data. Further information on the spatial distribution of a population in real time is required to improve exposure assessment [17].

An environmental health surveillance or tracking system may be defined as a system that performs the continuous collection, integration, analysis, and interpretation of data on human health effects relating to exposure to environmental hazards [18]. The potential for environmental health surveillance has previously been evaluated for the European population [19]. Exposure surveillance systems providing real-time exposure data may be developed based on air monitoring sensors and methods that may be used to evaluate an entire population within a region of interest [20].

To reduce the environmental risks, an environmental health surveillance system using a meaningful population exposure assessment methodology is required; this method can then evaluate the effectiveness of alternative policies and risk mitigation efforts. The purpose of this study is to present a methodology to assess air pollution exposure for an entire population by applying the latest technology. This includes sensor-based monitoring devices and the Internet of Things (IoT). This study also aims propose a plan to develop an environmental health surveillance system for human health risk management.

## 2. Materials and Methods

This review evaluates the latest published research on the air pollution exposure assessment for a population. Preference was given to reviewing recent studies presenting innovative approaches and new perspectives, particularly for methodologies and outcomes. Keywords related to exposure assessment were used as search criteria and their use in recent years was analyzed. The main search terms were filtered into journal categories representing subject areas such as “air pollutant”, “environmental health”, “exposure assessment”, “population exposure”, and “surveillance”. These terms were searched for alone and in combination using a range of electronic databases including PubMed, EBSCOW, ScienceDirect (Figure 1), Web of Science and Google Scholar. Among these articles, 102 were related with the search term. The full texts of the acquired articles were reviewed and filtered according to the inclusion and exclusion criteria. The exclusion criteria were as follows: studies written in a language other than English and studies including personal exposure. The acquired articles were reviewed and filtered according to the inclusion and exclusion criteria. The potentially eligible studies were selected and retrieved as full texts.

Exposure was conceptualized as the sum of the product of time spent by an individual or population in different microenvironments and the time-weighted average concentrations of air pollutants in those locations [9], as shown in the following equation [21].
(1)$$E=\int_{t0}^{t1}C\left(t\right)dt$$
where exposure (E) may be defined as a function of the air pollutant concentration (*C*) and time intervals (*t*).

## 3. Results

Exposure was calculated based on the sum of the time spent by individuals in different microenvironments and the average concentration of time-weighted pollutants at this specific location [9]. The general structure of the model with the environmental health surveillance system and the input dataset required to calculate exposure of the population and risk management is shown in Figure 2. Outdoor air pollutant concentrations were measured using sensor-based monitoring devices with a neighborhood spatial scale of 0.5 to 4.0 km [22]. Indoor air pollutant concentrations in houses, buildings, and transportation systems were measured or modeled by accounting for indoor sources and ventilation between the indoors and outdoors. The exposure scenario was generated by combining indoor and outdoor concentrations with the time-based activity patterns using a smartphone [23,24]. Data mining was used to process the measured or modeled concentrations of air pollutants in various microenvironments [25]. Then, these data were transported by wireless networks and web services, to accumulate a database on a server, and then expressed as web-access data [26]. The air pollution exposure for a population was analyzed in terms of the prevalence rate of access to the National Health Insurance Service. Based on this assessment, environmental health policies may be developed for risk management.

### 3.1. Sensor-Based Air Monitoring and Internet of Things Technology

Fixed air monitoring stations are limited in their capacity to provide air pollutant levels and the extent of exposure for the individual or population as they only provide data for a few locations and are expensive to operate [27]. As such, they cannot provide an assessment of human exposure, even if they supply accurate air pollutant concentrations [28,29]. Air quality in indoor and outdoor environments varies on a relatively small scale as the concentration of air pollutants in a particular location is largely dependent on local emission sources and air flow conditions [30]. Typically, air flow in urban environments is turbulent and difficult to predict even with sophisticated numerical modeling. This makes it difficult to assess the actual pollutant exposure levels for a population [31,32].

A means to assess pollutant exposure for a population based on air quality measurements may be the application of low-cost monitoring devices across a wide range of areas for environmental health surveillance. These methods are able to provide low-quality air pollutant concentration data, and may be used concurrently across a wide range of areas, offering high-resolution exposure assessment mapping in urban environments [11].

The need for mobile applications and a greater coverage of area is cost effective and may only be achieved by reducing the size and cost of portable sensor monitoring devices [33]. Recent commercial low-cost sensors represent an opportunity to establish an air pollutant measuring network that is able to monitor large areas with high spatial resolution at a lower cost than reference measurement methods. According to the Air Sensor Guidebook published by the United States Environmental Protection Agency (USEPA), there are various studies that have been conducted using low-cost air monitoring sensors that are currently being commercialized [34].

Recently, the combination of the IoT and environmental monitoring has become a new domain in environmental health due to the advancement of information and communication technologies, such as wireless fidelity (WiFi), long-term evolution (LTE), and other wireless communication technologies [35,36,37,38]. Although the use of sensor-based monitoring devices in exposure assessment is still controversial, the application of low-cost sensors has already shifted paradigms in air pollution monitoring and exposure assessment. As such, the application of these technologies will continue to grow, including supplementing atmospheric surveillance networks and expanding communication with the community [39,40].

### 3.2. Indoor Air Exposure

The measurement of outdoor air pollutants alone is insufficient to assess the exposure of a population as most individuals spend the majority of their time indoors. Dias and Tchepel [41] suggested that the spatio-temporal variability of urban air pollution, as well as indoor exposure and time–activity patterns, should be measured to assess exposure at the individual scale. The indoor environment is important, as it represents the environment in which individuals spend approximately 90% of their time [42,43,44]. Therefore, indoor air pollution may have a higher explanatory power than its outdoor counterpart in the evaluation of air pollution exposure for a population [45]. Despite the importance of indoor air exposure, the level of indoor air pollution measurement has been relatively insufficient compared to outdoor air pollution measurements; this is because it is challenging to measure air quality in private spaces.

The is major concern in terms of the reliability and accuracy of methods to estimate indoor air pollution levels [46]. The increasing interest in indoor–outdoor air quality relationships has led to the development of various techniques to study indoor source emissions, and the air exchange between outdoor and indoor pollutants. Existing standardized methods are insufficient due to the complex emission and dispersion of indoor air pollutants and site conditions [47]. As indoor air pollutants may be affected by multiple factors such as indoor sources, ventilation, decay, building type, and human activity, the limitations of small sample sizes usually produce inconsistent conclusions.

The impact of outdoor fine dust on the indoor environment is particularly important in many developing countries where outdoor fine dust pollution has been increasing [48]. According to Ji and Zhao [49], the contribution of outdoor PM_2.5_ concentrations to indoor PM_2.5_ concentrations was estimated at 54–96% (n = 90). There was a significant correlation between indoor and outdoor PM_2.5_ concentrations (*p* < 0.05) with a penetration factor of 0.21. Outdoor PM_2.5_ concentrations contributed approximately 52% and 42% to indoor PM_2.5_ concentrations in the cool and hot seasons, respectively [50].

Indoor air pollutant concentrations may be estimated using an indoor–outdoor (I/O) ratio, indoor air quality model, a statistical model, and artificial intelligence such as machine learning [51,52]. The I/O ratio may generally be used to estimate the concentration of indoor air pollutants [47]. However, the I/O ratio for PM_2.5_ concentrations was 0.12–3.36 with significant variation, as it may be affected by multiple factors [53]. According to Zuo et al. [49], the mean I/O ratio was estimated to be 0.73 ± 0.54, based on sensor monitoring in 4403 indoor air monitoring locations in Beijing over one year. Indoor pollutant concentrations in houses, buildings, and transportation systems may be measured or modeled through deduction from outdoor concentrations by applying I/O ratios; however, this method does not account for indoor air pollutant sources. To address this, other techniques have been proposed, such as indoor sources, generation and ventilation-based modeling, and data-based artificial intelligence [54].

### 3.3. Exposure Scenario Using Time–Activity Patterns

The modeling of the exposure scenario of air pollutants is carried out using various factors, such as air pollutant concentrations in microenvironments, geographical information of individuals based on time–activity patterns, and building characteristics [55,56]. While Breen et al. used [10] an exposure model (EMI) for individuals to conduct exposure assessments, the USEPA developed the air pollutants exposure model (APEX) to estimate exposure to PM_2.5_ [57]. Valari et al. [9] proposed exposure to atmospheric pollution modeling (EXPLUME), a local-scale individual exposure model that includes spatial activity event sequences and the infiltration of outdoor air pollutants into the indoor environment.

The exposure scenario for a population may be estimated by assessing exposures from different exposure scenarios for sub-populations [58]. The population may be classified into a subset of groups based on socio-demographic characteristics and time–activity patterns that produce similar exposure scenarios; this includes pre-school children, school students, housewives, office workers, and the elderly. The pollutant level exposure of the population may be estimated by integrating their exposure scenarios [42,44]. A new methodology was tested in Madrid, Spain to improve the estimation of population dynamics. The population exposure to NO_2_ during working days was assessed, and the results were compared with those obtained through census-based methodologies [59].

Tracking individual activity patterns is necessary to characterize the duration of exposure and properties of pollutants, as well as their time and location [60]. Various tools have been used in research to track the spatio-temporal mobilities of individuals in relation to activity tracking. These include the Global Positioning System (GPS) [61], WiFi network [24], and accelerometers [62]. The most common characteristic shared by activity tracking tools is the use of a mobile device, particularly smartphones. A GPS-based microenvironment tracker, known as MicroTrac, was developed by the USEPA to estimate the time spent in eight microenvironments using GPS data and geocoded building boundaries [63].

Many studies have reported on population activities and mobile patterns based on GPS and mobile phone applications [23,64]. Although the main advantage of data produced in these studies is the provision of high spatio-temporal resolution, their limitations are usually associated with small sample sizes. The user-centered mobile model approach demonstrates the potential to integrate mobile phone data in air quality management and epidemiological studies in order to classify a population in terms of the type of activities at home, at work, for leisure, and for travel [65]. However, these studies did not differentiate between individuals staying indoors and outdoors, flagging a potential to overestimate or underestimate exposure, given that most individuals spend the majority of their time indoors.

### 3.4. Big Data Mining and Exposure Distribution

Due to the significant increase in data volume, the application of big data analysis has gained global attention. Big data analytics is the process that assists organizations in developing more informed policy by collecting, organizing, and analyzing large quantities of data to search for hidden patterns, unknown correlations, trends, and other useful information [66]. As such, the integration and application of big data analysis is the future for environmental health and an area that urgently requires further development [67].

According to Zuo et al. [49], big data are able to provide a methodology to reduce heterogeneity in indoor PM_2.5_ exposure. By using the machine learning approach, Zheng et al. [32] developed a U-Air system that combines different types of heterogeneous big data to estimate air quality, such as meteorology, traffic flow, human mobility, road network structure, and point of interest. Recently, many researchers have begun to use the big data analysis approach because of the development of big data applications and the availability of environmental detection networks and sensor data [68]. Air quality has been estimated by a deep learning and image-based model [69].

Atmospheric dispersion and community multiscale air quality (CMAQ) models generally use computer-based simulations to calculate exposure distributions using the approximate spatial distribution of atmospheric pollutant concentrations. However, these types of models are limited as they must have accurate information on emissions, weather data, and the structure and geographical data of the region; this creates challenges in providing an exposure distribution [35,70].

Some researchers have investigated the spatial distribution of air pollutant concentrations from the geo-statistics perspective based on actual observations. This is due to the increasing number of fixed air monitoring stations and the greater availability of low-cost sensors for continuous spatio-temporal air quality monitoring [71,72,73]. For example, a novel application based on the optimal linear data fusion method was applied in combination with the kriging interpolation technique for data fusion between different types of PM_2.5_ sensors [35]. In recent studies, interpolation techniques such as land use regression (LUR), the inverse distance weighted (IDW) method, and the geo-statistical kriging algorithm have been widely used [31,74]. According to Berrocal et al. [75], the down-scaler model and universal kriging demonstrated better predictive performance than machine learning algorithms.

As a result, the average daily exposure distribution of air pollutants for a population has been calculated by the time-weighted average using exposure concentrations and time spent in each microenvironment to estimate population exposure [76]. That is, the total daily exposure (concentration (ppm or µg/m^3^) × time (h) × number of people) of a population may be indicated by accounting for the estimated concentration and the amount of real-time, dynamic population data based on mobile phones within a standard grid, such as 500 × 500 m and 1 × 1 km [59].

### 3.5. Environmental Health Surveillance System

Environmental health surveillance includes the collection of systematic exposure information on specific health effects that impact a population, the analysis and interpretation of such data, and effective data delivery to public health professionals and policymakers [77]. The need for an exposure surveillance system critical to prevent and control environmental diseases is increasing [18,78]. Given the importance of the environmental health surveillance system, the US Center for Disease Control and Prevention introduced the National Environmental Public Health Tracking Network System in 2010 [79]. The UK Health Protection Agency has been developing the Environmental Public Health Surveillance System [80], whilst the European population was also inventoried and Europe has evaluated the potential of environmental health surveillance [19].

Environmental health surveillance systems are able to illustrate the causal pathway from hazard to exposure to disease. In particular, an exposure surveillance system that provides real-time exposure data for air pollutants may be developed based on the characteristics of air monitoring sensors and the assessment of an entire population [20]. Public health policymakers may use insights from an environmental health surveillance system to promote public health, reduce exposure, and more accurately prevent the occurrence of diseases in efficient and cost-effective ways. An environmental health surveillance system for sustainable and healthy outcomes and an international network for practitioners and researchers that are able to monitor and use these systems to support countries and regions have been provided.

## 4. Discussion

As health effects caused by exposure to air pollution are a worldwide concern, public health policymakers are trying to reduce the exposure to air pollutants. It is necessary to identify the exposure distribution of air pollutants for an entire population [81,82]. Traditional methods are used to assess the air pollutant exposure of a population by collecting data from outdoor fixed air monitoring stations, and assign them to the home address of an individual using atmospheric dispersion models and spatial interpolation techniques [40,41,83]. However, the determination of a population’s exposure to air pollutants using measurements from fixed air monitoring stations may be insufficient in terms of their spatio-temporal resolution [83]. With the development of modern industrial society, sources of pollution and environmental change are becoming increasingly complex and the factors required to be accommodated in the model are increasing; this has led to a rise in calculation costs, uncertainty, and a decline in model accuracy [35].

This study aimed to improve the assessment of air pollution exposure for a population by overcoming some of these limitations. It also proposed a novel methodology to assess population exposure to air pollutants. According to Abelsohn et al. [84], using PM_2.5_ as an example, environmental health surveillance is divided into four categories; hazard, exposure, health effects, and intervention options. Among them, detailed information on individual exposure through the course of a human life is generally insufficient and the weakest source of information; this is despite the fact that exposure assessment is the most important link in describing the hazard–exposure–disease pathway [85].

Currently, measuring air pollutants using fixed air monitoring stations has limitations. However, this paradigm is changing with the application of low-cost, easy-to-use portable air pollution monitors, such as sensors that provide real-time data. These properties provide an opportunity to improve existing air pollution monitoring capabilities. It may also provide avenues to new air monitoring applications for population exposure to air pollutants [86,87].

Air pollution monitoring has already changed the paradigm with the application of low-cost sensor-based monitoring, and the application of these technologies will continue to grow. In particular, current low-cost sensing technologies complement routine ambient air monitoring networks, and expand the communication of risk to communities. This alone heralds a paradigm shift in air quality monitoring, which was mostly implemented by the Ministry of Environment in the past. An additional paradigm shift is the increased use of artificial intelligence or other advanced data processing approaches to improve sensor-based monitoring in agreement with reference monitors [39].

Although individuals spend more time indoors, exposure assessment and the health effects of indoor air pollution have not been studied as extensively as outdoor air pollution. One of the main reasons is that the measurement of indoor air quality is not easy as indoor environments are typically private; this means there is a lack of information on indoor air quality pollution [88]. As indoor air quality varies from time to time due to changes in building conditions, human activity, and weather conditions, short-term sampling cannot account for all types of variation. This highlights the need for long-term monitoring, which may be resolved by sensor-based monitoring instruments [89].

Under the framework of the IoT and ICT, the air pollutant exposure of the population may be analyzed systematically by continuously collecting device information and environmental big data distributed from different time and space points. Data mining and spatio-temporal data analysis techniques can be used to extract valuable information from environmental big data; this information may be offered to governments for policymaking or further academic analysis [35,90].

Exposure estimation methods may be inadequate, as they do not address the spatio-temporal and exposure variabilities inherent in a population. In addition, these estimation methods neglect indoor air pollution; by far the largest source of exposure to air pollution for a population [61]. Exposure may be defined as a function of concentration and time. As such, tracking the activity patterns at an individual and sub-population scale should be considered. The total daily exposure of a population with concentration, time, and the number of people may be generated by estimated concentrations and dynamic population data using smartphones within a standard grid. Whether individuals stay indoors or outdoors at that time should be evaluated [91]. The use of personal information may be a problem for phone owners, telecom operators, researchers, and the public. However, dynamic population data may be used for public purposes because personal information and identifying mobile data that may directly be linked to individuals has been removed [92].

## 5. Conclusions

This study proposed a framework to evaluate population exposure to environmental pollutants and the development of an environmental health surveillance system. Through the use of this framework, the application of recent technologies to evaluate population exposure to air pollutants may allow policymakers to formulate informed, evidence-based decisions on risk assessment. Population-based exposure distribution is useful to understand population-specific differences in risk and identify priorities for environmental health intervention. The exposure data for a population obtained from this methodology may be used in air quality management, risk management, and environmental health policy development. It may also be used in epidemiological research to study correlations with specific diseases.

## Figures and Tables

**Figure 1 toxics-08-00074-f001:**
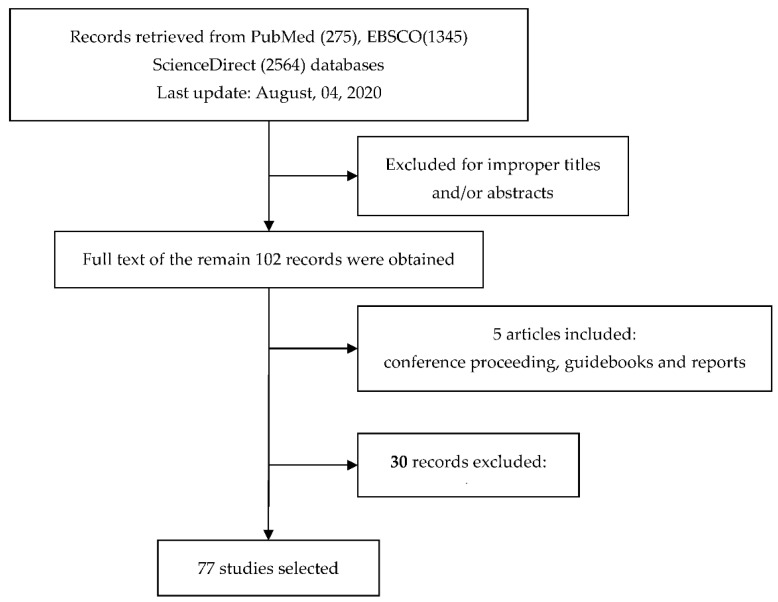
Flow diagram of the systematic review.

**Figure 2 toxics-08-00074-f002:**
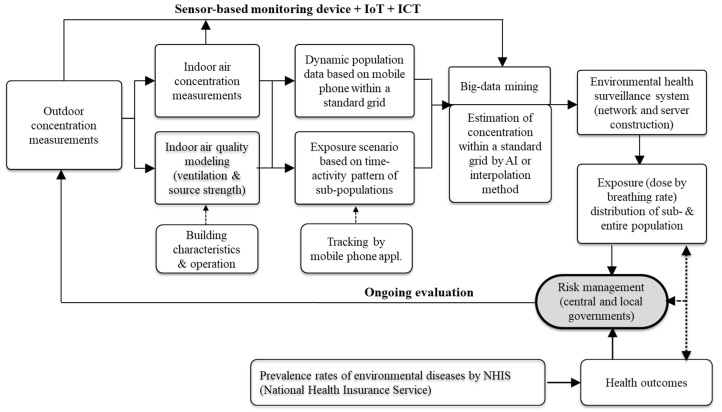
Overview of the development of an environmental health surveillance system for a population through the advancement of air pollution exposure assessment.

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
