# Peer review of "Environmental Health Surveillance System for a Population Using Advanced Exposure Assessment"

_toxics, 2020, doi:10.3390/toxics8030074_

Round 1

Reviewer 1 Report

This interesting paper presents a scoping literature review of methods to estimate air pollution exposure to populations.  I call it a scoping review, even though the authors do not, because it addresses the existing literature and summarizes the need for better ways to estimate air pollution such that it can be extrapolated to individual exposures.  So, this paper is not a research study as much as it is a review.

Given the above, the authors are encouraged to use the PRISMA methodology to conduct this review.  It would entail a formal description of the methods used to search the literature, the methods used to identify relevant studies and judge  their quality, etc. 

I note that the substantive part of such a review is already present in the paper, just the underlying methods to get there need further elaboration.

Author Response

Thank you for your thoughtful review. We greatly appreciate your very good comments.

Comments and Suggestions for Authors:

1. This interesting paper presents a scoping literature review of methods to estimate air pollution exposure to populations. I call it a scoping review, even though the authors do not, because it addresses the existing literature and summarizes the need for better ways to estimate air pollution such that it can be extrapolated to individual exposures. So, this paper is not a research study as much as it is a review.

  • We again appreciate the encouraging comments. As for your comments and suggestions, this is review manuscript. This study suggested a methodology to develop an environmental surveillance system that evaluates the distribution of air pollution exposures for a population within a target area.

2. Given the above, the authors are encouraged to use the PRISMA methodology to conduct this review. It would entail a formal description of the methods used to search the literature, the methods used to identify relevant studies and judge their quality, etc.

  • This manuscript is a literature review. However, it is a review of the contents of past papers to propose a new methodology in future exposure assessment. Therefore, it is somewhat different from meta-analysis that produces new results from these papers. In other words, the exposure assessment proposed in this manuscript is not abundant in the past papers related to the new methodology, so it can be said that it has limitations in collecting and considering systematic literature under PRIMA methodology. Therefore, considering these limitations, the authors obtained as much as possible the existing reported papers that are believed to be consistent with subject of this manuscript, classified and reviewed them according to what the authors wanted to assert. Flow diagram of the systematic review was inserted as shown in Figure 1.

3. I note that the substantive part of such a review is already present in the paper, just the underlying methods to get there need further elaboration.

  • As your suggestion, we revised the Material and Methods section.

Reviewer 2 Report

The content of the manuscript focuses on a very important issue, which is the assessment of exposure to indoor air pollution and the use of advanced technologies for this purpose.

Despite the fact that it is a review article, which has already been indicated in the title of the paper, the Authors propose a certain methodology in this regard, aimed at constructing an "environmental health surveillance system". However in section 1, 2 or 3 it would be useful to refer to this proposal more specifically, including highlighting its possible innovative elements.

This type of system is based on low-cost air quality sensors. Although the Authors are aware of the low quality of data generated by this type of sensors (line 139), the paper (e.g. sections 4 and 5) should pay more attention to the need for their improvement (in addition to what is stated on lines 315-317). The reliability of the measurement results and the conclusions drawn based on them depends, among others, on the accuracy of these sensors. Thus, the statement contained in lines 232-324 could be reinforced with a slightly larger number of works presenting verified examples of their applications, if they are known.

Editorial comments:

On lines 148 and 191 there is an unnecessary dot before the references (before [34] and [54], respectively).

The font size of the headings of sections 3.1-3.5 should be made uniform.

On line 307 there should probably be "monitors" instead of "minotirs".

At the end of the sentence contained in lines 328-330 there are unnecessarily two dots.

In the References section, the correctness of specifying items [44] and [55] should be checked - unnecessary digits “1” at the beginning and incorrect formatting of the journal name (it should be Italic) and the year (it should be bold).

Author Response

Thank you for your thoughtful review. We greatly appreciate your very good comments.

Comments and Suggestions for Authors:

1. The content of the manuscript focuses on a very important issue, which is the assessment of exposure to indoor air pollution and the use of advanced technologies for this purpose

  • We again appreciate the encouraging comments.

2. Despite the fact that it is a review article, which has already been indicated in the title of the paper, the Authors propose a certain methodology in this regard, aimed at constructing an "environmental health surveillance system". However in section 1, 2 or 3 it would be useful to refer to this proposal more specifically, including highlighting its possible innovative elements.

  • As your suggestion, we revised the section 1, 2 and 3. In order to propose the exposure assessment method for a population, we have revised them sequentially based on the processes shown in Figure 2.

3. This type of system is based on low-cost air quality sensors. Although the Authors are aware of the low quality of data generated by this type of sensors (line 139), the paper (e.g. sections 4 and 5) should pay more attention to the need for their improvement (in addition to what is stated on lines 315-317). The reliability of the measurement results and the conclusions drawn based on them depends, among others, on the accuracy of these sensors. Thus, the statement contained in lines 232-324 could be reinforced with a slightly larger number of works presenting verified examples of their applications, if they are known..

  1.  
  • We fully agree with the limitations on the accuracy of the current low-cost air quality sensor measurements. However, as described in the Discussion section, the sensor technology continues to grow. And an additional paradigm shift is the increased use of artificial intelligence or other advanced data processing approaches to improve sensor-based monitoring in agreement with reference monitors.

4. Editorial comments:

1) On lines 148 and 191 there is an unnecessary dot before the references (before [34] and [54], respectively).

  • We deleted an unnecessary dot.

2) The font size of the headings of sections 3.1-3.5 should be made uniform

  • We revised the font size of the headings of sections 3.1-3.5.  

3) On line 307 there should probably be "monitors" instead of "minotirs".

  • Right. We changed the “monitirs” into “monitors”.  

4) At the end of the sentence contained in lines 328-330 there are unnecessarily two dots.

  • We revised and deleted an unnecessary dot.

5) In the References section, the correctness of specifying items [44] and [55] should be checked - unnecessary digits “1” at the beginning and incorrect formatting of the journal name (it should be Italic) and the year (it should be bold).

  • We deleted the unnecessary digits “1” and revised the incorrect formatting of the journal name and the year.

Round 2

Reviewer 1 Report

The authors have addressed my comments.